# Correction of Breech Presentation with Moxibustion and Acupuncture: A Systematic Review and Meta-Analysis

**DOI:** 10.3390/healthcare9060619

**Published:** 2021-05-22

**Authors:** Jian-An Liao, Shih-Chieh Shao, Chian-Ting Chang, Pony Yee-Chee Chai, Kok-Loon Owang, Tse-Hung Huang, Chung-Han Yang, Tsai-Jen Lee, Yung-Chih Chen

**Affiliations:** 1Department of Traditional Chinese Medicine, Keelung Chang Gung Memorial Hospital, Keelung 204, Taiwan; frank771124@cgmh.org.tw (J.-A.L.); kchuang@cgmh.org.tw (T.-H.H.); cs8336@gmail.com (T.-J.L.); 2Department of Pharmacy, Keelung Chang Gung Memorial Hospital, Keelung 204, Taiwan; s.c.shao@hotmail.com (S.-C.S.); rrrctc@cgmh.org.tw (C.-T.C.); ponychai@cgmh.org.tw (P.Y.-C.C.); thenight_eve@hotmail.com (K.-L.O.); 3School of Pharmacy, Institute of Clinical Pharmacy and Pharmaceutical Sciences, College of Medicine, National Cheng Kung University, Tainan 701, Taiwan; 4Center of Evidence-Based Medicine, Keelung Chang Gung Memorial Hospital, Keelung 204, Taiwan; dr3939@gmail.com; 5Division of Rheumatology, Allergy, and Immunology, Department of Internal Medicine, Linkou Chang Gung Memorial Hospital, Taoyuan 333, Taiwan; 6Division of General Internal Medicine, Department of Internal Medicine, Keelung Chang Gung Memorial Hospital, Keelung 204, Taiwan

**Keywords:** breech, pregnancy, maternal and prenatal health, moxibustion, acupuncture, systematic review, meta-analysis

## Abstract

Acupuncture-type interventions (such as moxibustion and acupuncture) at Bladder 67 (BL67, Zhiyin point) have been proposed to have positive effects on breech presentation. The aim of this systematic review and meta-analysis was to evaluate the effectiveness and safety of moxibustion and acupuncture in correcting breech presentation. We searched PubMed, MEDLINE, Embase, the Cochrane Central Register of Controlled Trials (CENTRAL), the Chinese Electronic Periodical Services (CEPS), and databases at ClinicalTrials.gov to identify relevant randomized controlled trials (RCTs). In this study, sixteen RCTs involving 2555 participants were included. Compared to control, moxibustion significantly increased cephalic presentation at birth (RR = 1.39; 95% CI = 1.21–1.58). Moxibustion also seemed to elicit better clinical outcomes in the Asian population (RR = 1.42; 95% CI = 1.21–1.67) than in the non-Asian population (RR = 1.20; 95% CI = 1.01–1.43). The effects of acupuncture on correcting breech presentation after sensitivity analysis were inconsistent relative to control. The effect of moxibustion plus acupuncture was synergistic for correcting breech presentation (RR = 1.53; 95% CI = 1.26–1.86) in one RCT. Our findings suggest that moxibustion therapy has positive effects on correcting breech presentation, especially in the Asian population.

## 1. Introduction

Breech presentation is a common malposition in the third trimester of pregnancy. The frequency of breech presentation in term pregnancies is 3%–4% in America and approximately 2% in China [1,2]. Risk factors for breech presentation include preterm labor, uterine anomaly, multiparity, placenta previa, and polyhydramnios [3]. Serious complications, such as traumatic injuries or asphyxia, can occur during vaginal delivery [4]. Therefore, a planned Caesarean section is recommended for pregnant women with breech presentation at childbirth [3]. However, Caesarean section is not free from complications, including wound infection, adhesions, hemorrhage, or scar rupture during subsequent labor [5]. Some non-invasive therapies are available, including knee-chest position management and external cephalic version (ECV). However, there is insufficient evidence to support knee-chest position management, and ECV is a painful procedure for pregnant women [6,7]. 

Moxibustion and acupuncture have a long history in the treatment of various problems, including fetal malposition. The interventions are similar because they both stimulate acupoints to achieve a therapeutic effect. Moxibustion is a traditional Chinese procedure that utilizes the heat generated from a burning moxa stick (made from herbal preparations containing *Artemisia vulgaris*) to stimulate acupuncture points [8,9]. Several clinical trials have shown that moxibustion at Bladder 67 (BL67), also known as the Zhiyin point, elicits positive effects on breech presentation without serious adverse events [10,11]. However, systematic reviews and meta-analysis have reported conflicting results regarding the effects of moxibustion on breech presentation. For example, Vas et al. [12] and Li et al. [13] reported that moxibustion has positive effects on non-vertex presentation. However, Coyle et al. [14] suggested that moxibustion treatment may not improve non-cephalic presentations at birth relative to no treatment. To determine the efficacy of moxibustion on breech presentation, additional clinical trials since 2012 have investigated the effects of moxibustion [15,16]. Acupuncture has also been reported to correct fetal malposition, although evidence from systematic reviews and meta-analysis is lacking [17]. To fill this research gap, we conducted an updated systematic review and meta-analysis to evaluate the effects and safety of these acupuncture-type interventions in correcting breech presentation.

## 2. Materials and Methods

This systematic review and meta-analysis study are reported in accordance with the statement of preferred reporting items for systematic reviews and meta-analysis (PRISMA). The protocol was registered on PROSPERO with a registration number: CRD42020192572.

### 2.1. Search Strategy 

In this systematic review, we included all RCTs on the use of acupuncture-type interventions (i.e., moxibustion and acupuncture) in the management of breech presentation, regardless of whether the RCTs were blinded. We performed literature searches in PubMed, MEDLINE, Embase, the Cochrane Central Register of Controlled Trials (CENTRAL), the Chinese Electronic Periodical Services (CEPS), and databases at ClinicalTrials.gov from the inception of the source to 31 January 2021. Keywords for literature search included “breech,” “labor presentation,” “acupuncture,” “electroacupuncture,” “acupressure,” and “moxibustion.” We explored our literature search with MeSH headings without restrictions of language, publication type, or date. We applied a filter to narrow the number of articles which fit the specific study type (e.g., RCTs) and study question (e.g., intervention). The details of the search strategy are presented in Appendix A (Appendix A). We also searched the reference lists of included studies and related articles in PubMed and clinical trial databases to identify relevant RCTs.

### 2.2. Study Selection and Data Extraction

We selected eligible studies based on the following inclusion criteria: (1) the study was an RCT; (2) pregnant women in the 28th–35th week of gestation with a normal pregnancy and an ultrasound diagnosis of breech (non-vertex) presentation were included; (3) the interventions consisted of moxibustion alone, traditional acupuncture or electro-acupuncture alone, or moxibustion and acupuncture; (4) comparisons between interventions and control measures (e.g., observation, usual care, or knee-chest position) were conducted; and (5) outcome measures (e.g., fetal presentation at birth and adverse events) were reported. Studies were excluded based on the following criteria: (1) the study was non-randomized, quasi-experimental, observational, qualitative, or did not involve human subjects; (2) had wrong or no comparators; or (3) had incomplete outcome data. Two authors independently selected articles according to the inclusion and exclusion criteria by screening the titles, abstracts, and full texts of included studies. We extracted the following information from studies that met the inclusion criteria: study characteristics (e.g., author, publication year, study design and settings, inclusion/exclusion criteria, methods of randomization), participant characteristics (age, gender, co-morbidities), interventions (types, duration), comparisons (types of control groups) and outcomes (types of outcome measures, adverse events). We retrieved data from individual studies having an intention-to-treat principle. Any disagreements about whether to include a study were resolved by a third reviewer. 

### 2.3. Assessment of the Risk of Bias in Included Studies

Two authors independently assessed the methodological quality of each included clinical trial, according to the Cochrane risk of bias tool for randomized controlled trials (RoB 2.0) [18]. RoB 2.0 is composed of five domains, including bias arising from the randomization process (allocation), bias due to deviations from the intended interventions (performance), bias due to missing outcome data (follow-up), bias in the outcome measurement (measurement), and bias in the selection of the reported results (reporting). The authors rated each domain as either low risk, some concerns (uncertain risk of bias), or high risk. Discrepancies were resolved by the third reviewer.

### 2.4. Data Synthesis and Statistical Analysis 

We compared moxibustion with control, acupuncture with control, and moxibustion plus acupuncture with control. The primary outcome was the fetal presentation at birth, and the secondary outcome was adverse events. 

Data were analyzed using Review Manager Software (version 5.3.5). Dichotomous outcomes were extracted from each study to compute the RR with a 95% CI. The pooled RR and the associated 95% CI were estimated by the Mantel-Haenszel method. Numbers needed to treat (NNT) were calculated from the formula (NNT = 1/absolute risk reduction). We assessed clinical heterogeneity by comparing the methodologies and study designs of the included studies. Statistical heterogeneity of effect sizes between studies was assessed using the I^2^ statistic and Q statistic with an X^2^ test. We defined statistical heterogeneity using *p* ≤ 0.1 for the X^2^ test or I^2^ ≥ 50%. In the meta-analysis, a fixed-effect model was performed when there was no significant heterogeneity, and a random-effects model was performed when the heterogeneity was significant. A funnel plot was produced to detect possible publication bias. Sensitivity analysis was performed to test the robustness of results by excluding trials that used low-quality methodologies. To assess between-group differences and explain heterogeneity, we carried out a subgroup analysis. Because regional differences may exist, we reported treatment effects on breech presentation separately.

## 3. Results

### 3.1. Study Selection and Characteristics

We identified 198 studies using our search strategy and included 16 studies based on our inclusion and exclusion criteria. We summarize the process of study identification and selection in Figure 1, and present the characteristics of each of the included studies in Table 1. All included studies were randomized controlled trials. The size of the study populations ranged from 20 to 406 persons. The 16 studies included a total of 2555 participants; eight studies included participants from China [10,11,19,20,21,22,23,24], two studies included participants from Italy [25,26], and the others included participants from France [27], Australia [28], Switzerland [29], Croatia [17], Denmark [16], and Spain [15]. Most studies were published in English (56.3%); others were published in Chinese (37.5%) and French (6.2%). 

Regarding study interventions, 13 RCTs compared moxibustion with control [10,11,15,16,19,21,22,23,24,25,27,28,29]; two RCTs compared acupuncture with control [17,22]; two RCTs compared moxibustion plus acupuncture with control [20,26]. Treatment was applied to BL67 in all included studies. The duration of each intervention was distinct from study to study. The typical application time of moxibustion or acupuncture was 15–20 min. The treatment sessions were typically conducted over 7–14 days. The gestational age was different among studies, ranging from 28 to 37 weeks, when acupuncture or/and moxibustion was performed.

### 3.2. Methodological Quality of Included Studies 

The risk of bias for included studies is shown in Figure 2 and Figure 3. All studies were assessed as having low or uncertain levels of risk of bias, except in the domains of allocation and follow-up. We present the details of the risk of bias assessment in Appendix A (Appendix A). In general, the quality was moderate in all included studies, except for four studies [20,22,23,25] that were assessed as having a high risk of bias in the domain of either allocation or follow-up.

### 3.3. Efficacy of Interventions

The meta-analysis of the included studies revealed a beneficial effect of acupuncture-type interventions on correcting breech presentation at delivery (average RR = 1.45; 95% CI = 1.28–1.65; random effect model, I^2^ = 66%) (Figure 4). The forest plot was divided into three subgroups: (1) moxibustion, (2) acupuncture, and (3) moxibustion plus acupuncture.

Fetal presentation was investigated in the results of 13 studies with 2063 participants comparing moxibustion with control [10,11,15,16,19,21,22,23,24,25,27,28,29]. The pooled data show a significant increase in cephalic presentation at birth (RR = 1.39; 95% CI = 1.21–1.58; random effect model, I2 = 64%). The NNT is 6 (95% CI = 4–11).

Two clinical trials involving 146 patients compared acupuncture with control [17,22]. The meta-analysis reveals no differences between treatment and control groups (RR = 2.78; 95% CI = 0.84–9.19; random effect model, I2 = 85%). 

Pooled data from two trials with 346 participants reveals significant difference between moxibustion plus acupuncture and control groups in the meta-analysis [20,26] (RR = 1.53; 95% CI = 1.26–1.86; random effect model, I2 = 0%). The NNT is 5 (95% CI = 3–9).

### 3.4. Sensitivity Analysis: Excluding Four Trials with a High Risk of Bias

Limiting the meta-analysis to the 12 trials with moderate to low risk of bias [10,11,15,16,17,19,21,24,26,27,28,29] which investigated the effects of acupuncture-type interventions including moxibustion, acupuncture, and moxibustion plus acupuncture reveal significant effects on correcting fetal malposition (RR = 1.36; 95% CI = 1.23–1.51; random effect model, I^2^ = 41%) (Figure 5). The sensitivity analysis for the moxibustion subgroup reveals a result like that of the previous analysis (RR = 1.34; 95% CI = 1.19–1.51; random effect model, I^2^ = 47%). The NNT is 7 (95% CI = 5–12). Only one trial that evaluated acupuncture versus control shows more cephalic presentation in the acupuncture group (RR = 1.68; 95% CI = 1.11–2.55). The NNT is 4 (95% CI = 2–20). Only one trial reports that moxibustion plus acupuncture had more cephalic presentation relative to control (RR = 1.42; 95% CI = 1.06–1.90). The NNT is 7 (95% CI = 3–45).

### 3.5. Subgroup Analysis of Moxibustion

Moxibustion is effective in the Asian population (RR = 1.42; 95% CI = 1.21–1.67; random effect model, I^2^ = 71%) and in the non-Asian population (RR = 1.20; 95% CI = 1.01–1.43; random effect model, I^2^ = 0%) (Figure 6).

### 3.6. Adverse Events

Information on adverse events was presented in four trials. Because of the clinical heterogeneity between the included studies, we did not perform a meta-analysis of adverse events. Cardini et al. in 2005 reported adverse events (41.5%) related to moxibustion [25]. Patients had abdominal pain, throat problems, and unpleasant odor with or without nausea. Cardini et al. in 1998 and Vas et al. reported that no adverse events occurred in the moxibustion or control groups [10,15]. Neri et al. observed no adverse effects on participants who received moxibustion plus acupuncture or usual care [26].

### 3.7. Publication Bias

We used Review Manager Software (Version 5.3.5) to evaluate the publication bias. The sample size of most studies was >100 participants with two comparison arms except for Do 2011 [28], Li 1996 [22], and Millereau 2009 [27]. Funnel plots are typically symmetrical for studies with large sample sizes (Figure 7). However, for studies with small sample sizes, no study reported a negative result, which suggests that publication bias is probable in the literature reporting correction of breech presentation with moxibustion and acupuncture.

## 4. Discussion

Our study found that acupuncture-type interventions (including moxibustion, acupuncture, and moxibustion plus acupuncture) at BL67 increase the frequency of cephalic presentation at birth. Moxibustion seemed to be more effective in correcting non-vertex presentation in the Asian population than in the non-Asian population.

Previously, Vas et al. found that moxibustion had positive effects on correcting non-vertex presentation, although they noted that there was considerable heterogeneity among studies [12]. Li et al. demonstrated that moxibustion was effective in correcting breech presentation, but non-randomized controlled trials were included in this study [13]. The results of these two studies differed from those of Coyle et al. [14], who found that moxibustion did not reduce the frequency of non-cephalic presentation relative to no treatment [14]. This discrepancy could be attributed to emerging clinical trials in recent years. In addition, Coyle et al. did not include all relevant trials, such as Chen, 2007 [21], Do, 2011 [28], Li, 1996 [22], Millereau, 2009 [27], and Yang, 2008 [19]. Our study included only RCTs that were eligible and up-to-date.

To minimalize the impact of potential bias, a sensitivity analysis was performed; such an analysis was not reported as being conducted in most previous studies. After comparing the net effects of different acupuncture-type interventions before and after sensitivity analysis, a positive effect on correcting breech presentation, particularly with moxibustion alone or in combination with acupuncture, is consistent. Our findings provide robust support of the effectiveness of moxibustion on correcting breech presentation. 

The mechanism of moxibustion is not fully understood. Moxibustion at BL67 is thought to stimulate the production of prostaglandin and estrogen, which increases uterus contractions that lead to fetal movements [30,31]. Traditional Chinese medicine (TCM) theory teaches that disharmony of qi and blood may cause fetal malposition. It is thought that moxibustion at BL67 tonifies Yang qi and dredges channels to correct fetal position [19,21].

Some studies suggest that the effects of treatment might be related to ethnicity [32,33]. We performed a subgroup analysis to assess differences between ethnic groups and found that moxibustion seemed to be more effective in correcting non-vertex presentation in Asians than in non-Asian populations. To the authors’ best knowledge, this is the first article that investigates the effect of moxibustion on breech presentation in different races. However, the mechanism of this phenomenon is unclear.

During pregnancy, acupuncture has been hypothesized to have beneficial effects on pelvic pain or labor pain [34,35]. In TCM theory, moxibustion or acupuncture applied at BL67 is thought to activate blood circulation and dredge channels to correct fetal malposition [20]. However, there have been few studies on the use of acupuncture to treat breech presentation, and there has been no systematic review or meta-analysis in the literature to date. In our study, only two clinical trials were retrieved and included in the meta-analysis, but the risk of bias in one of those trials [22] was rated as “high.” The result of the subsequent sensitivity analysis revealed that the effect of acupuncture was inconsistent. Therefore, reports on the effects of acupuncture should be interpreted with caution.

According to Coyle et al., there was a positive effect on breech presentation using moxibustion combined with acupuncture [14]. Nevertheless, only one trial was included in the meta-analysis. Our study included a new trial [20], and the result was similar. The pooled RR of moxibustion versus moxibustion plus acupuncture was 1.39 vs. 1.53 without analysis and 1.34 vs. 1.42 with sensitivity analysis. The combination of moxibustion and acupuncture appears to exert a synergistic effect on correcting breech presentation.

Previous systematic reviews included RCTs with different controls, including knee-chest position or observation [12,13]. However, one systematic review by Hofmeyr et al. found that there was no difference in cephalic presentation between knee-chest position and observation [7]. Therefore, we included clinical trials with no-effect controls, including knee-chest position and observation.

Moxibustion and acupuncture are generally safe when administrated by experienced clinicians, and both are less expensive than Caesarean section in general practice. In a study by Ineke et al., moxibustion reduced the number of Caesarean sections performed in pregnant woman with breech presentation and was cost-effective when compared to expectant management [36]. A previous study pointed out that there were no significant differences in the comparison of moxibustion with usual care, with respect to premature births or premature rupture of the membranes [12]. We performed meta-analysis of these two outcomes, and had similar results (see Appendix A Appendix A). Because the use of TCM theories is increasing in many countries, the modality of using moxibustion might be more widely deemed as being beneficial in obstetric patients.

This study was limited in several aspects. First, there might be publication bias in the meta-analysis. Second, the sample sizes of some included studies were too small for RCT design. Finally, the application time of treatment (15–20 min) and treatment duration (7–14 days) differed between studies

## 5. Conclusions

Our updated systematic review and meta-analysis suggested that moxibustion has a positive effect on correcting breech presentation. However, more randomized, controlled clinical trials are needed to evaluate whether our estimate of the magnitude of the effect of moxibustion remains constant.

## Figures and Tables

**Figure 1 healthcare-09-00619-f001:**
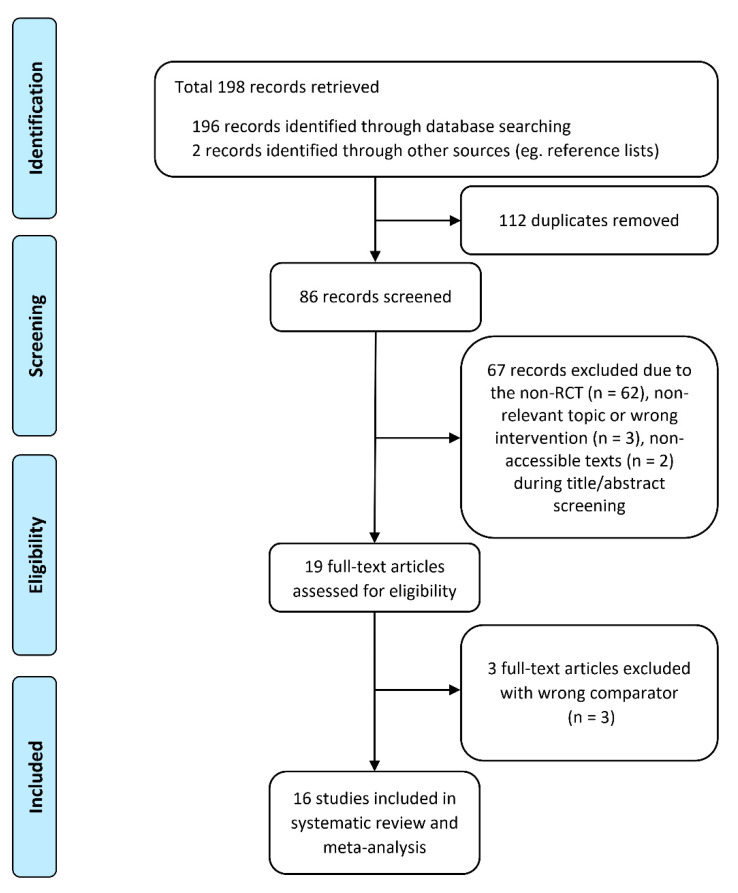
Flow chart of the identification and selection of studies for inclusion.

**Figure 2 healthcare-09-00619-f002:**
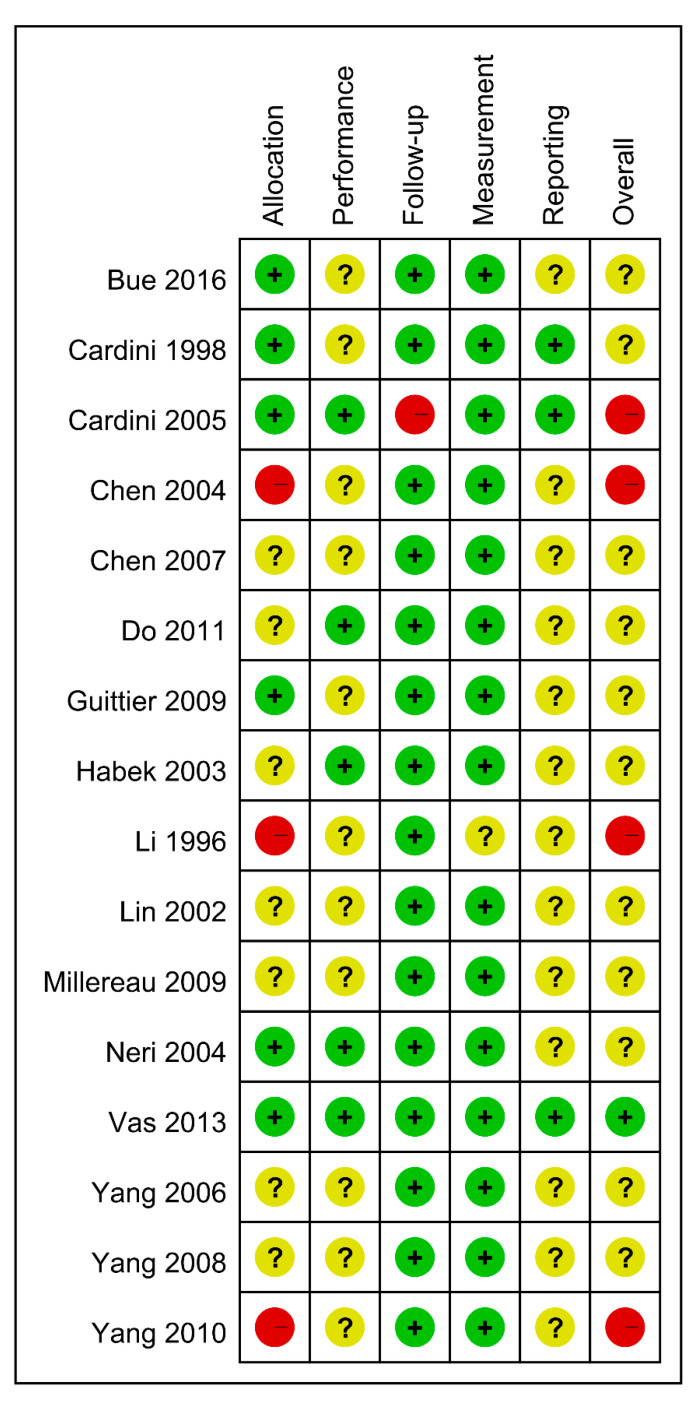
Methodological quality summary: reviews authors’ judgement on each methodological quality item for each included study.

**Figure 3 healthcare-09-00619-f003:**
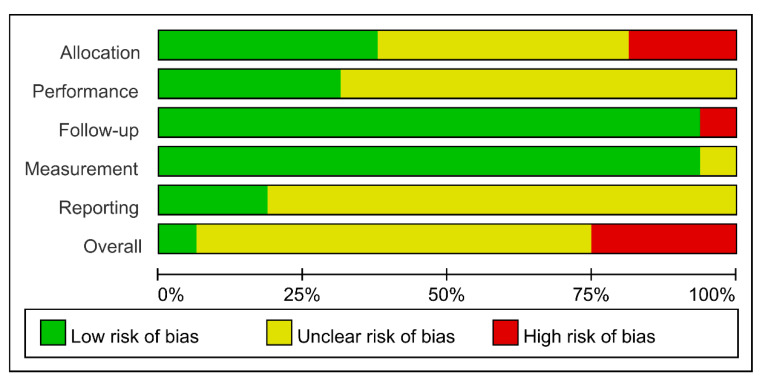
Methodological quality graph: reviews authors’ judgement on each methodological quality item presented as percentage for each included study.

**Figure 4 healthcare-09-00619-f004:**
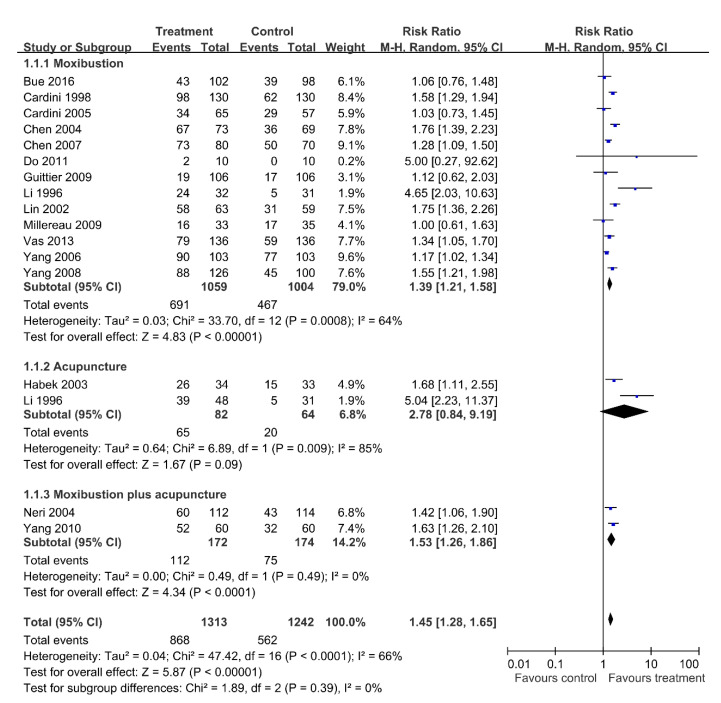
Forest plot of each comparison: Acupuncture-type interventions versus Control; Outcome: Cephalic presentation.

**Figure 5 healthcare-09-00619-f005:**
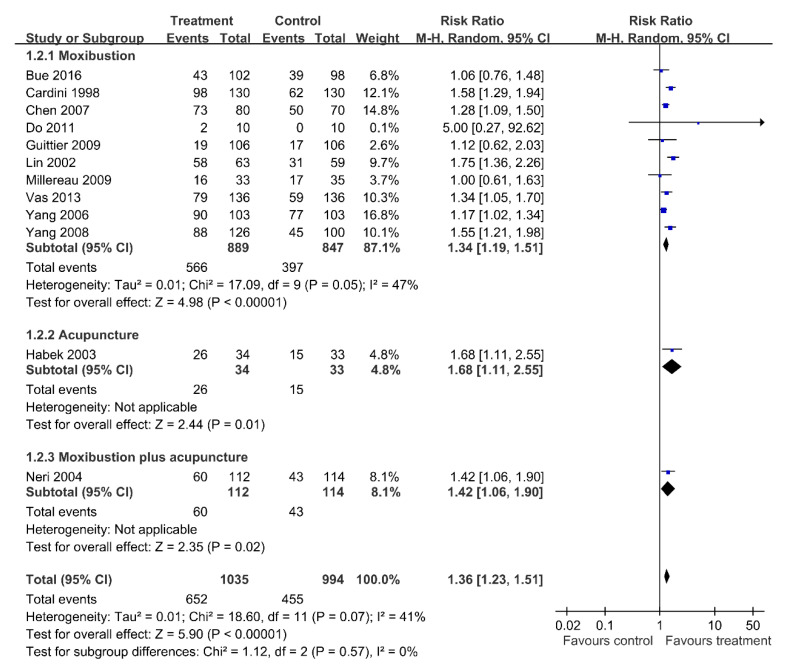
Sensitivity analysis: Acupuncture-type interventions versus Control; Outcome: Cephalic presentation.

**Figure 6 healthcare-09-00619-f006:**
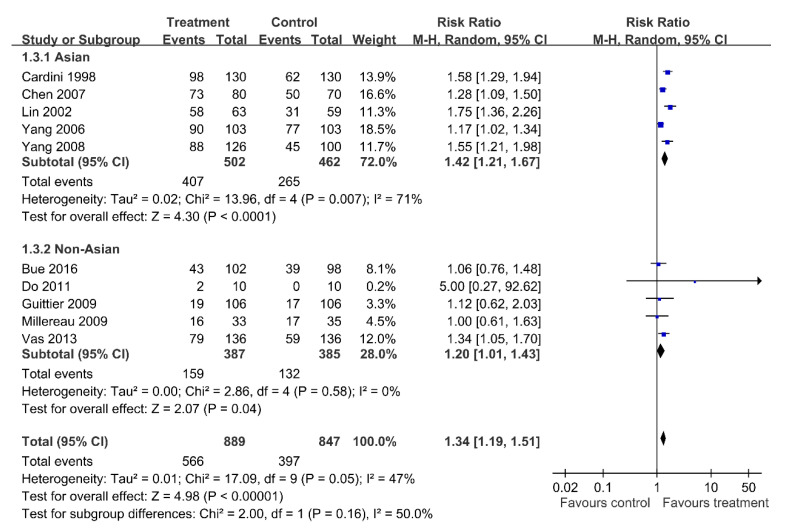
Subgroup analysis: Moxibustion versus Control; Outcome: Cephalic presentation.

**Figure 7 healthcare-09-00619-f007:**
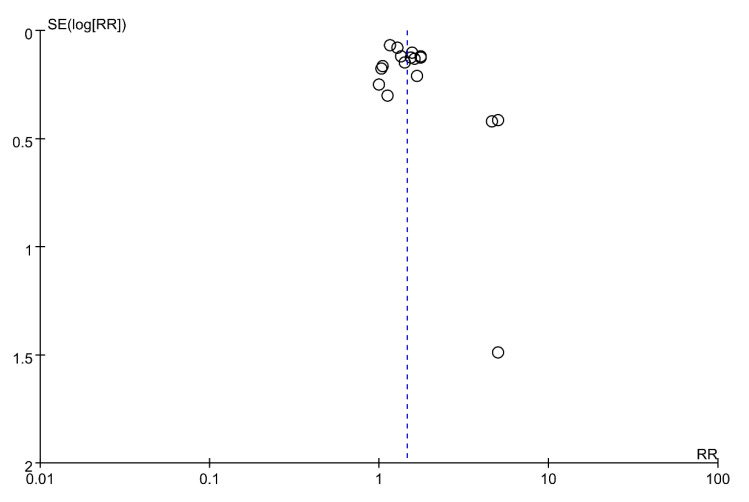
Funnel plot of the studies included in the meta-analysis.

**Table 1 healthcare-09-00619-t001:** Characteristics of the included studies.

Study	Study Design	Patient Population	Intervention	Control	Outcomes
Bue 2016 [16]	RCT	200	Moxibustion at BL67; daily for 15–20 mins, preferably in the evening; 14 days	Observation	Cephalic presentation; rate of ECV and Cesarean delivery
Cardini 1998 [10]	RCT	260	Moxibustion at bilateral BL67; 15 mins on each side; 7 days	Observation	Cephalic presentation; Cesarean delivery rate
Cardini 2005 [25]	RCT	123	Moxibustion at BL67; 15 mins; twice daily; 7 days	Observation	Cephalic presentation
Chen 2004 [23]	RCT	142	Moxibustion at BL67; twice daily, 10–15 mins each time; up to 14 days	Observation and knee-chest position	Cephalic presentation
Chen 2007 [21]	RCT	150	Moxibustion at bilateral BL67; 20 mins a day; up to 15 days	Knee-chest position	Cephalic presentation
Do 2011 [28]	RCT	20	Moxibustion at bilateral BL67; 10 mins on each side; 10 days	Observation	Cephalic presentation
Guittier 2009 [29]	RCT	212	Moxibustion at bilateral BL67; 10 mins on each side; 14 days	Observation	Cephalic presentation
Habek 2003 [17]	RCT	67	Acupuncture at BL67; 30 mins; twice a week	Observation	Cephalic presentation; Cesarean delivery rate
Li 1996 [22]	RCT	111	Electro-acupuncture at bilateral BL67; 30 mins; 6 days Moxibustion at bilateral BL67; 20 mins; 6 days	Observation	Cephalic presentation
Lin 2002 [24]	RCT	122	Moxibustion at bilateral BL67; twice daily for 20 mins	Knee-chest position	Cephalic presentation
Millereau 2009 [27]	RCT	68	Moxibustion at BL67; 15–20 mins; 7 days	Observation	Cephalic presentation
Neri 2004 [26]	RCT	240	Moxibustion plus acupuncture at bilateral BL 67; total 40 mins; twice a week; 14 days	Observation	Cephalic presentation; Cesarean section rate
Vas 2013 [15]	RCT	406	Moxibustion at bilateral BL67; 20 mins; 14 days	Observation	Cephalic presentation; Cesarean delivery rate
Yang 2006 [11]	RCT	206	Moxibustion at BL67; 15–20 mins; 7 days	Knee-chest position	Cephalic presentation
Yang 2008 [19]	RCT	226	Moxibustion at bilateral BL67; 15 mins; up to 14 days	Knee-chest position	Cephalic presentation
Yang 2010 [20]	RCT	120	Moxibustion plus acupuncture at bilateral BL67; once daily; 14 days	Knee-chest position	Cephalic presentation

BL67: Bladder67 (Zhiyin point), ECV: external cephalic version, RCT: randomized controlled trials.

## Data Availability

Not applicable.

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
