# Peer review of "Correction of Breech Presentation with Moxibustion and Acupuncture: A Systematic Review and Meta-Analysis"

_healthcare, 2021, doi:10.3390/healthcare9060619_

Round 1

Reviewer 1 Report

This is an interesting review which evaluated the effectiveness and safety of moxibustion and acupuncture in correcting breech presentation. I would like to ask you two questions.

In result part, are there any RCTs which presented the difference of RR according to the gestational week when acupuncture or/and moxibustion was performed?

In Discussion part, it is written that moxibustion in BL67 increases uterus contractions that lead to fetal movements. Increasing uterus contractions may lead premature delivery. Are there any discussion on safe gestational week when to perform acupuncture or/and moxibustion? 

Reviewer 2 Report

Dear authors,

Thank you very much for this interesting article that may be applicable to certain patient population and provider. 

Although I personally do not have enough experience with these type of practices, the results that are analyzed by you and the conclusions derived, seem correct to me. I have no comments to add and wish you luck in the publication process

Best regards

Reviewer 

Reviewer 3 Report

This work presents a systematic review and meta-analysis on the correction of breech presentation by moxibustion and acupuncture either alone or in combination. Overall, the work is methodologically sound and the conclusions sound. Some minor comments: 1. Line 42 – why is it necessary to use the abbreviation CS? It seems that this is only used on the following line. 2. Line 143, 157 and figures – the Figures are misnumbered in both text and in the figures themselves
